# Natural Evolution of Porcine Epidemic Diarrhea Viruses Isolated from Maternally Immunized Piglets

**DOI:** 10.3390/ani13111766

**Published:** 2023-05-26

**Authors:** Yufang Ge, Feiyang Jiang, Sibei Wang, Heqiong Wu, Yuan Liu, Bin Wang, Wei Hou, Xiuju Yu, Haidong Wang

**Affiliations:** 1College of Veterinary Medicine, Shanxi Agricultural University, Jinzhong 030801, China; 2Single Molecule Nanometry Laboratory (Sinmolab), Nanjing Agricultural University, Nanjing 210095, China

**Keywords:** porcine epidemic diarrhea virus, immunological evolution, natural recombinant

## Abstract

**Simple Summary:**

During the long-term co-evolution of the virus and the host, even closely related vaccines may emerge with incomplete protective immunity due to the mutations or deletions of amino acids at specific antigenic sites. The mutation of PEDV was accelerated by the recombination of different strains and the mutation of the strains adapting to the environment. These mutations either cause immune escape from conventional vaccines or affect the virulence of the virus. Therefore, researching and developing new vaccines with cross-protection through continuous monitoring, isolation and sequencing are important to determine whether their genetic characteristics are changed and to evaluate the protective efficacy of current vaccines.

**Abstract:**

The porcine epidemic diarrhea virus (PEDV) can cause severe piglet diarrhea or death in some herds. Genetic recombination and mutation facilitate the continuous evolution of the virus (PEDV), posing a great challenge for the prevention and control of porcine epidemic diarrhea (PED). Disease materials of piglets with PEDV vaccination failure in some areas of Shanxi, Henan and Hebei provinces of China were collected and examined to understand the prevalence and evolutionary characteristics of PEDV in these areas. Forty-seven suspicious disease materials from different litters on different farms were tested by multiplex PCR and screened by hematoxylin-eosin staining and immunohistochemistry. PEDV showed a positivity rate of 42.6%, infecting the small and large intestine and mesenteric lymph node tissues. The isolated strains infected Vero, PK-15 and Marc-145 multihost cells and exhibited low viral titers in all three cell types, as indicated by their growth kinetic curves. Possible putative recombination events in the isolates were identified by RDP4.0 software. Sequencing and phylogenetic analysis showed that compared with the classical vaccine strain, PEDV SX6 contains new insertion and mutations in the S region and belongs to genotype GIIa. Meanwhile, ORF3 has the complete amino acid sequence with aa80 mutated wild strains, compared to vaccine strains CV777, AJ1102, AJ1102-R and LW/L. These results will contribute to the development of new PEDV vaccines based on prevalent wild strains for the prevention and control of PED in China.

## 1. Introduction

The GIIb non-S-INDEL porcine epidemic diarrhea (PED) virus (PEDV) strain that appeared in China in 2010 caused a large-scale outbreak of PED [1,2,3] and brought substantial economic losses to the farming industry. PEDV can cause infection in pigs of all ages and induce severe clinical symptoms of acute diarrhea, vomiting, dehydration and high mortality in newborn piglets. The virus mainly colonizes the epithelial cells of the small intestinal mucosa. It causes intestinal villi atrophy and shedding by destroying small intestinal epithelial cells during the initial stage of infection [4]. In recent years, the genetic diversity of PEDV strains in China has changed due to their continuous variation. The most prevalent variants of PEDV in Henan, Shanxi and other provinces are the GIIa and GIIb subgroups [5,6,7,8].

PEDV belongs to the *Alphacoronavirus* genus of the coronavirus family, with a genome size of approximately 28 kb and consisting of seven open reading frames (ORFs) [9]. The S gene is a structural gene of PEDV and is one of the hypervariable regions in the genome [10]. Its monomer consists of a β-folded S1 subunit and a S2 subunit consisting of a series of discontinuous α-helices [11]. The S1 subunit binds to the cell receptor, and the HR1 and HR2 regions of the S2 subunit then form a 6-helix bundle (6-HB) to mediate the fusion of virus and cell membrane [12]. ORF3 is the only accessorial gene of PEDV. In general, wild-type ORF3 has a full length of 675 nt, and cell-adapted or attenuated strains such as classical strains CV777, DR13 and P-5V have a 49 nt or 51 nt deletion [13]. This deletion can be used as a distinguishing marker between wild and attenuated strains. Full-length and truncated ORF3 could interact with S protein to regulate virus replication and affect virus virulence [14].

Vaccination is an effective method to prevent and control epidemic diseases [15]. In China, 36 producers have received approval for producing porcine diarrhea vaccines. PEDV vaccines are mainly live and inactivated vaccines in divalent and trivalent combinations with porcine infectious gastroenteritis, and a PEDV-inactivated vaccine (strain XJ-DB2) was developed by Tiankang Bio in 2022. Current vaccine strains mainly include CV777, AJ1102, ZJ08, SCSZ-1, AJ1102-R and LW/L strains, while ZJ/15, WN-R and HB17 strains have entered clinical trials. During the long-term co-evolution of the virus and the host, even closely related vaccines may emerge with incomplete protective immunity due to the mutations or deletions of amino acids at specific antigenic sites. Therefore, understanding the evolutionary diversity of viruses is necessary to evaluate the potential role of vaccines in driving PEDV evolution.

## 2. Materials and Methods

### 2.1. Collection of Cell Culture and Clinical Samples

Vero, PK-15, Marc-145 and BHK21 cells were grown in dulbecco’s modified eagle medium (DMEM, Cytiva, Marlborough, MA, USA) containing 6% inactivated fetal bovine serum (Invitrogen, Australia) and a mix of penicillin (at a working concentration of 100 U/mL) and streptomycin (at a working concentration of 0.1 mg/mL). The cells were cultured at 37 °C and 5% CO_2_ and then used for the test after the monolayer was full.

Small intestines, large intestines, mesenteric lymph nodes and feces were collected from piglets that died or had clinical signs within 10 days of birth in 2019–2021 from a total of 13 pig farms in Henan, Hebei and Shanxi provinces of China. Vaccine use in suspected samples collected in Shanxi, Henan and Hebei are shown in Table 1. About 1–2 g of small intestine or feces were obtained and homogenized with phosphate buffer solution (PBS) at 1:3 ratio, frozen and thawed repeatedly for three times, and centrifugated at 8000 g and 4 °C for 15 min. The supernatant was collected, filtered using a disposable filter with 0.22 μm pore diameter, and stored at −80 °C preservation as the virus liquid for the follow-up test.

### 2.2. Multiple Detection of Virus

In brief, 250 μL of virus solution was mixed with 750 μL of RNAiso Plus (Takara, Beijing, China), lysed, centrifugally washed, and dried for 5–7 min. The precipitate was dissolved in 20 μL of DEPC-treated water. The following primer sequences were designed and synthesized (Universal, Beijing, China), including PEDV, porcine transmissible gastroenteritis virus (TGEV) and porcine rotavirus (PoRV). PEDV-F 5′-CATATGTTTGTAATGGTAACTCTCGT-3′, PEDV-R 5′-AGAGCAAGATAATTGAGT CTAGCT-3′; TGEV-F 5′-GATATGTTTGTAATGGCAACCCT-3′, TGEV-R 5′-CTCTATAG CTGAACGATACTTACGT-3′; and PoRV-F 5′-TTTACTCTACATAAAGCATCAAT-3′, PoRV-R 5′-GACGGCAACTCAACCTCTCACAT-3′. The fragments of PEDV, TGEV, and PoRV were amplified using FastKing one-step RT-PCR (General biol, Chuzhou, China) under the following reaction conditions: 42 °C for 30 min; 94 °C for 4 min; 94 °C for 30 s, 53 °C for 30 s and 72 °C for 60 s for 35 cycles; 72 °C for 5 min; and 4 °C temporary storage. The amplification products were detected with 1% agarose gels and positive samples were used for subsequent tests.

### 2.3. Histological Analysis of the Materials

The collected small intestine, large intestine and mesenteric lymph nodes from the pigs with the isolate PEDV were fixed with 4% paraformaldehyde, dehydrated with graded ethanol and xylene transparent. The tissues were embedded in paraffin, cut into 6 μm sections, and stained with hematoxylin and eosin. Ethanol fractionated dewaxing and antigen retrieval were performed. SP Rabbit & Mouse Hrp Kit (DAB) (Shiji kangwei, Taizhou, China) and neutral resin patch were used for immunohistochemical observation.

Interventionary studies involving animals or humans, and other studies that require ethical approval, must list the authority that provided approval and the corresponding ethical approval code.

### 2.4. Sequence Analysis and Phylogenetic Construction

Seven samples were selected for sequence amplification and sequencing, including E6, 1-1, 1 + 2, 7.1, ZMD5, 4-1 and SX6. Viral RNA was extracted and reverse transcribed into cDNA using HiScript II Q RT SuperMix for qPCR Kit (Vazyme, Nanjing, China). The S1, S2 and ORF3 genes of PEDV were amplified using this template. S1-F 5′-ATGAAGTCTTTA ACCTACTTCTGGT-3′, S1-R 5′-AATACTCATACTAAAGTTGGTGGGA-3′; S2-F 5′-ATGAGGACAGAATATTTAC AG-3′, S2-R 5′-CTGCACGTGGACCTTTTC-3′; and ORF3-F 5′-ATGTTTCTTGGACTTTT TCAATACACG-3′, ORF3-R 5′-TCATTCACTAAT TGTAGCATACTCGT-3′. PCR products were separated by agarose gel electrophoresis and purified by agarose gel DNA recovery kit. The target fragments were ligated to pMDTM19-T Vector using Takara pMDTM19-T Vector Cloning Kit (Dalian, China). Blue and white spots were screened, and the positive clones were selected and sent to BGI for Sanger sequencing to confirm the correct transformation. Sequence splicing was performed using bioinformatics primer 5, DNAman and DNAstar software, version 17.4.2. Eighteen PEDV reference strains containing different PEDV genotypes (Ia, Ib, IIa and IIb) were obtained from the GenBank database for phylogenetic analysis, as shown in Table 2. Cluster W comparison in MEGA7.0 software and the Neighbor-Joining method were used to construct the phylogenetic tree. Node value was calculated by bootstrap repeated 1000 times to calculate node value. The amino acid sequences of the S and ORF3 genes of the commercial vaccine strains were homologated with DNAstar software.

### 2.5. Virus Isolation and Purification

The cells were inoculated in T25 flask and cultured in an incubator at 37 °C and 5% CO_2_ to form a monolayer with the addition of serum-free maintenance medium containing 0.3% tryptone phosphate broth. The cells were treated with 0.25% trypsin at 0, 0.5, 1, 2, 5, 10, 15 and 20 μg/mL and observed for 48 h to select the appropriate enzyme concentration. The cells were passaged, the old medium was discarded, and the cells were washed with PBS. The cells were incubated with 1 mL of virus stock solution and appropriate cell maintenance solution, incubated at 37 °C for 2–3 h, and added with new cell maintenance solution. The cells were cultured at 37 °C and 5% CO_2_ until 80–90% of CPE appeared, frozen–thawed three times, and stored at −80 °C for the next passage. Unfortunately, the only strain that produced CPE was SX6.

The cells were seeded in six-well plates, and the virus solution was diluted to 10^−1^–10^−3^ by 10-fold gradient and adsorbed at 37 °C for 1–2 h. In brief, 3% low melting point agarose solution and cell maintenance solution were mixed into each well in a ratio of 1:1. After cooling and solidification, the cells were inverted and cultured in an incubator at 37 °C and 5% CO_2_. Afterward, 0.002% neutral red stain was incorporated for 1.5–2 h when pathological changes appeared. If CPE was not evident, then the cells were incubated for a further 12–24 h to pick out the right size plaque spots. By cloning 3 times in this way, the virus was purified.

### 2.6. Selection of Susceptible Cells

PK-15, Marc-145, Vero and BHK21 cells were seeded into six-well plates, treated with appropriate trysin concentrations of 1, 2, 5 and 1 μg/mL, and observed for 48 h. The cells were then infected with 0.5 MOI. CPE was observed under a 100× microscope at 0, 12, 24 and 48 h.

### 2.7. Tissue Culture Infection Dose (TCID_50_) Measurement

The virus solution was diluted to 10^−1^–10^−8^ in a tenfold gradient, cultured in 96-well plates and washed with PBS. Afterward, 100 μL of maintenance solution was added, followed by 100 μL of dilution solution under the corresponding gradient. Eight well replicates were prepared for each gradient. As the control group, normal cells were added with 200 μL of maintenance solution and incubated at 37 °C and 5% CO_2_ in an incubator. The number of lesioned wells was observed after 4–5 days. TCID_50_ values were calculated using Reed–Muench method [16].

### 2.8. Growth Dynamic Curve of Different Cells Infected by the Virus

PK-15, Marc-145 and Vero cells were seeded into 96-well cell plates, and the supernatants of cytosol cultures were collected at 0, 6, 12,18, 24, 30, 36 and 42 h. The viral titers of PK-15, Marc-145 and Vero cells were detected by TCID_50_.

### 2.9. Indirect Immunofluorescence Assay (IFA)

Vero cells were seeded in 24-well plates and supplemented with 0.1 MOI virus solution, incubated for 48 h, and added with cell maintenance solution. Normal cells were used as negative control. The medium was discarded at 8, 24, and 48 h post-infection, washed three times with PBS, fixed with precooled 4% tissue cell fixative solution at 4 °C for 30 min, washed three times with PBS with 0.2% Triton X-100 (Solarbio, Beijing, China), and permeabilized at 37 °C for 20 min. The samples were washed three times with PBS and blocked in 5% goat serum at 37 °C for 1 h. The blocking solution was discarded, and the antiserum against PEDV as primary antibody was diluted at a ratio of 1:20 and incubated at 37 °C for 1 h. The samples were then washed three times with PBS, and Alexa Fluor 488 goat anti-mouse IgG secondary antibody (Bioss, Beijing, China) was diluted at a ratio 1:100 and incubated at 37 °C for 1 h in dark. The samples were washed in PBS three times, dropwise added to an anti-fluorescence decaying tablet (including DAPI), gently covered with a glass slide, and observed at 200× by a laser confocal microscope.

### 2.10. Analysis of S Gene Recombination

Possible recombination events of S gene in the isolates were identified by RDP4.0 recombination detection program. RDP, Chimaera, BOOTSCAN, GENECONV, 3Seq, MaxChi and SiScan programs embedded in RDP4.0 software package were applied for analysis.

## 3. Results

### 3.1. PEDV Virus Screening

The H&E staining of small intestine, large intestine and mesenteric lymph nodes of isolated strains showed that the villi of the small intestine became shorter and blunt, the lamina propria glands disappeared, the epithelial cells became slightly vacuolated, and the mucosal lamina propria was infiltrated by red blood cells. The intestinal gland of the large intestine atrophied, and the mucosal lamina propria was infiltrated by red blood cells. Part of the medulla of the lymph nodes was hyperchromatic and infiltrated by cortical red blood cells, and part of the lymphocyte was abnormal. IHC showed that PEDV was detected in the lymph nodes of the small intestine, large intestine and mesentery as shown in Figure 1A.

The primers were designed with reference to the gene sequences of PEDV, TGEV and PoRV. According to the results of one-step multiplex PCR amplification shown in Figure 1B, 20 of the 47 clinically suspicious samples tested were found to be PEDV positive, giving a PEDV positivity rate of 42.6%. Individual samples were labeled as mixed cases of PEDV and PoRV.

### 3.2. S, ORF3 Amino Acid Sequence Alignment and Evolutionary Analysis

Single positive samples of PEDV from different farms at different sampling locations were used as templates to amplify the S1, S2 and ORF3 genes. Samples amplified and sequenced included E6, 1-1, 1 + 2, 7.1, ZMD5, 4-1 and SX6.

The S1, S2 and ORF3 gene fragments of PEDV were obtained by Sanger sequencing, and S1 and S2 gene fragments were spliced to obtain a 4161 nucleotide S protein encoding 1387 amino acids. The S1 protein contains three domains: SP (aa 1–18), S1-NTD (aa 19–233) and COE (aa 499–638), whereas the S2 protein contains the following major structural domains: SS2 (aa 748–755), SS6 (aa 764–771), HR1 (aa 978–1117), HR2 (aa 1274–1313), TM (aa 1328–1350). As shown in Figure 2A, the amino acid sequence alignment of the S gene revealed 48–57 amino acid mutations in the seven sequenced samples compared to vaccine strain CV777. COE, SS2 and SS6 were the major neutral B-cell epitope regions of this structural domain. The amino acid YSNIGVCK in the SS2 region was conserved, and no mutations were observed. One amino acid substitution exists on SS6 and 8–12 mutant sites in the COE core region. In comparison with vaccine strains CV777, AJ1102, AJ1102-R and LW/L, the following mutations occurred in SX6, 7.1, 1-1 and 4-1 at positions 522 (A → S), 531 (L → V) and 541 (F → L), and it is not known whether these mutations affect viral neutralization activity. Most of the 32–35 amino acid mutations or insertions were concentrated in S1-NTD, with varying degrees of mutations in SP, HR1 and HR2. The results of the ORF3 amino acid sequence comparison are shown in Figure 2B. There are 8–20 amino acid mutations in ORF3 compared to the vaccine strain CV777.

The sequences of 18 reference strains were obtained from GenBank, and the amino acid sequences of S and ORF3 were compared for developmental relationships using the maximum likelihood method as shown in Figure 2C,D. On the basis of the S sequence, the reference strains CV777, LZC and SM98 represent gene type GIa, and DR13, AH-2018-HF1 and JS2008 represent GIb subgroups. The other 12 prevalent strains belong to variant subgroups GIIa and GIIb. The three strains in this study, ZMD5, E6 and 1 + 2, belong to the same GIIb as the YN90 strain identified in China after 2015. the other four strains, 4-1, 7.1, 1-1 and SX6, belong to the GIIa subgroups, as do some of the prevalent strains in China and PC22A identified in the USA in 2017. Based on S amino acid sequence homology matching, the sequenced samples showed 93.2–98.4% homology with the vaccine strains. The amino acid homology between the sequenced samples and the vaccine strain was 85.5%-100% concerning the ORF3 sequence. Three of the strains, ZMD5, E6 and 1 + 2, belong to group II with strain YN90 identified in China after 2015, while the other four strains, 4-1, 7.1, 1-1 and SX6, belong to group III with PC22A found in the USA in 2017.

### 3.3. Isolation and Identification of PEDV Virus

Seven samples from the sequencing were cultured for cell isolation. Samples with no CPE from the blind passages were discarded, and those with lesions were purified to obtain the new strain PEDV SX6.

PK-15, Marc-145, Vero and BHK21 cells were infected with 0.1 MOI. CPE was observed under a microscope at 100×. The results showed that the isolate was not infectious to BHK21 cells but was infectious to PK-15, Marc-145 and Vero cells, all of which developed lesions. At 12–18 h after infection, the cells began to gather and contract. Fusion and vacuolation were observed in PK-15 cells after 24 h infection. At 48 h after infection, the syncytial state appeared in large amounts. In addition, 100% CPE lesions appeared as shown in Figure 3A.

The isolate infected the PK-15, Marc-145 and Vero cells by 0.5 MOI inoculation, and the culture supernatant was collected at different time points. Virus titers were detected by TCID_50_ as shown in Figure 3B. At 42 h of infection, the assay titers were 10^6.2^ TCID_50_/mL, 10^5.8^ TCID_50_/mL and 10^5.8^ TCID_50_/mL, respectively. The virus expressed a slightly higher viral titer in PK-15 cells.

The isolate strain was inoculated into Vero cells, and the polyclonal antibody against PEDV was used as the primary antibody for IFA detection. As shown in the results of 200× immunofluorescence test in Figure 3C, green fluorescence could be observed in the cells infected with the virus. The fluorescence gradually increased with the duration of virus infection.

### 3.4. Analysis of S Gene Recombination

The hypothetical recombination events of the isolates PEDV SX6 were inferred with RDP4.0 software. Seven embedded detection programs, namely RDP, Chimaera, BOOTSCAN, GENECONV, 3SEQ, MaxChi and SiScan were used to identify the recombinant sequence and confirm Av. *p*-Val value. As shown in Figure 4, MISSOURI270/20 (KR265846.1) and HeN170821 (MK862249.1) were identified as parental strains, and six of the programs supported recombination event generation (GENECONV, Av. *p*-val = 1.49 × 10^−2^; bootscan, Av. *p*-val = 7.414 × 10^−3^; maxchi, Av. *p*-val = 4.955 × 10^−4^; Chimara, Av. *p*-val = 5.253 × 10^−5^; siscan, Av. *p*-val = 1.252 × 10^−4^; 3 seq, AV. *p*-val = 5.159 × 10^−4^). The sequence comparison of recombinant and nonrecombinant regions showed 98.7% and 99% sequence similarity for the potential major parent strain Missouri270/2014 (KR265846.1) and the minor parent strain HeN170821 (MK862249.1), respectively. The phylogenetic tree showed that the isolate belonged to the GIIa subgroup similarly to the major parent strain MISSOURI270/2014. Meanwhile, the minor parent strain HeN170821 belonged to the GIIb subgroup. This significant difference in evolution supported the possibility of recombination events in these strains.

## 4. Discussion

PEDV is causing damage to the global pig industry. Even though farms have standardized various doses of the PEDV vaccine, more PED cases are being reported. In Guangdong Province, China, the detection rate of positive PEDV was 47.0% in eight pig farms [17]. In Henan Province, 18 regional samples of pigs with diarrhea were collected, and the PEDV infection rate was as high as 51.65%. The highest PEDV positivity rate was found in lactating piglets (60.47%) and was widespread in both PEDV-vaccinated (25.00%) and non-vaccinated pigs (62.29%) [18]. We performed multiplex RT-PCR on PEDV clinical material collected from 13 pig farms from November 2019 to April 2021 from sows immunized with the PEDV vaccine whose piglets became sick or died. We found a PEDV positivity rate of 42.6%. One of the reasons for the high level of PEDV infection in lactating piglets is that the vaccine dose, vaccination interval and route of vaccination may all affect the immune effect of the vaccine [19]. Secondly, the available commercial vaccine strains do not provide complete immune protection. The low levels of PEDV-specific and IgA antibodies produced by sows may result in inadequate protective antibodies for lactating piglets [20].

Phylogenetic analysis based on S genes in this study showed that 4-1, 7.1, 1-1 and SX6 belonged to the GIIa subgroup, while ZMD5, E6 and 1 + 2 belonged to the GIIb subgroup. The researchers divided PEDV into the GI classical genome and the GII variant genome [21]. The GIIa and GIIb subgroups are non-S INDEL strains in the S-structural domain, while the GIIb subgroup is a highly virulent recombinant strain [22]. The evolution of PEDV and the continuous mutation of the viral genome have led to the emergence of new strains. Our sequencing results suggest that these samples from different farms may come from different origins. S gene variation is one of the main causes of vaccine immune failure [23]. The proteins encoded can induce the production of neutralizing antibodies against PEDV. The main neutralizing epitopes identified include COE, SS2 and SS6 [24,25]. In this trial, the amino acid sequence alignment of clinical samples with the S gene of vaccine strain CV777 revealed one amino acid substitution on SS6 and 8–12 mutations in the core region of COE. Not only are there four common amino acid substitutions in the COE region (522A → S, 554T → S, 599G → S and 610A → E), but there are also three other amino acid substitutions (526L → H, 528S → G and 532V → I). The S1-NTD concentrates 32–35 amino acid mutations, which may change the main conformation of the S1-NTD [7]. The above mutations in S1 can cause variation in the viral genome, and the amino acid changes in the encoded proteins may affect viral virulence and altered antigenicity [26]. 8–20 amino acid mutations were present in the ORF3 amino acid sequence comparison of seven clinical samples. ORF3 has been reported to contain four transmembrane structural domains: TMD1 (aa 40–63), TMD2 (aa 75–97), TMD3 (aa 116–139) and TMD4 (aa 150–173) [27]. Mutations detected in clinical samples in TMD1, TMD2 and TMD4 may impact virulence and need further investigation.

Many mutations in the S and ORF3 gene sequences of PEDV SX6 isolates, particularly in the S protein, may be determinants of its pathogenicity. These mutations may alter its antigenicity, pathogenicity and neutralization properties [28]. In addition to the accumulation of point mutations, homologous recombination between the same species is a common route of the genetic evolution of coronaviruses [17]. Analysis of S gene recombination in this strain has shown that intra-subgroup recombination may occur at loci 1-1071, with potential recombination breakpoints. It has been shown that recombination in PEDV can occur between and within subgroups [29], and thus recombinant strains exhibit evolutionary diversity. Frequent recombination events may lead to viruses evading the host’s immune defenses, compromising immune efficacy.

The virulence of prevalent strains of PEDV is influenced by a variety of genes, of which the S gene is only one necessary determinant [30]. The accessory protein encoded by the ORF3 gene is also an important virulence factor for the virus in the natural host [31,32]. In this study, PEDV SX6 was able to form CPE on PK-15, Marc-145 and Vero cells, with viral titers up to 10^6.2^ TCID_50_/mL, 10^5.8^ TCID_50_/mL and 10^5.8^ TCID_50_/mL at 42 h of infection in vitro. We have not yet determined the pathogenicity of the PEDV isolate SX6 in this study, and it would be worthwhile investigating further whether this isolate could be a safe and effective candidate strain.

## 5. Conclusions

The mutation of PEDV was accelerated by the recombination of different strains and the mutation of the strains adapting to the environment. These mutations either cause immune escape from conventional vaccines or affect the virulence of the virus. Therefore, researching and developing new vaccines with cross-protection through continuous monitoring, isolation and sequencing are important to determine whether their genetic characteristics are changed and to evaluate the protective efficacy of current vaccines.

## Figures and Tables

**Figure 1 animals-13-01766-f001:**
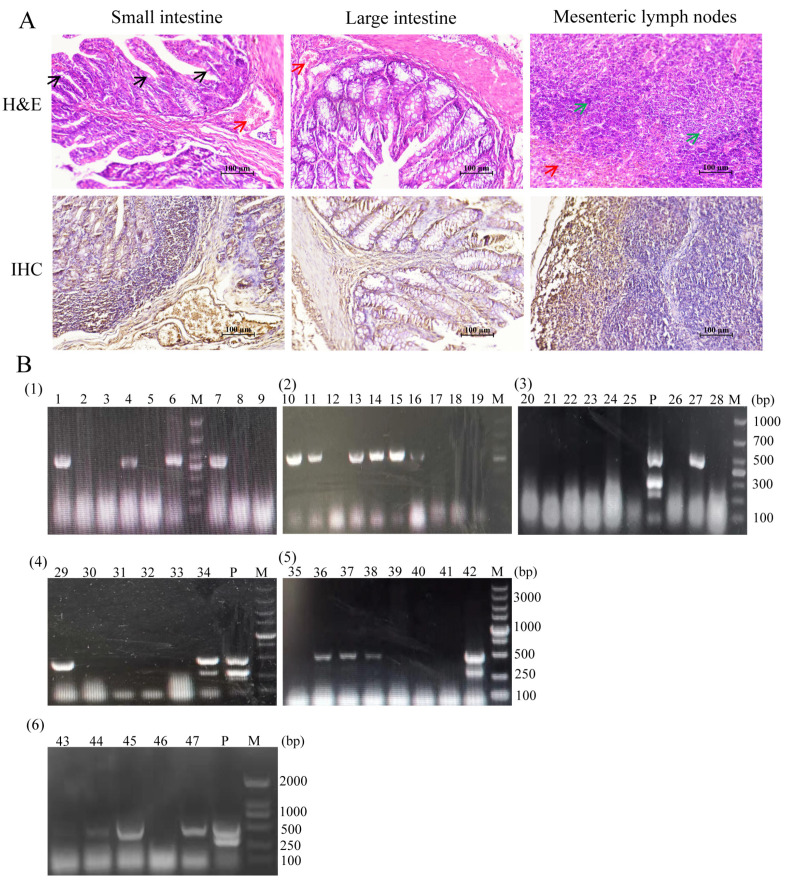
Morphological characteristics of the diseased tissues and multiplex detection of the samples. (**A**) Small intestine, large intestine and mesenteric lymph nodes were processed for H&E and IHC. Red arrows indicate erythrocyte infiltration, black arrows indicate mild vacuolization of intestinal epithelial cells, and green arrows indicate abnormal lymphocyte morphology. IHC shows infection with PEDV antigen in different tissues with brown staining. (**B**) Multiplex results of suspicious samples. M: DNA Ladder, 1–47: suspicious sample code, P: vaccine strain as control. The amplified fragments of PEDV, TGEV and PoRV were 532, 299 and 416 bp, respectively. (**1**)–(**3**) Sampling time November 2019–January 2020, (**4**)–(**5**) sampling time November 2020–December 2020, (**6**) sampling time March–April 2021.

**Figure 2 animals-13-01766-f002:**
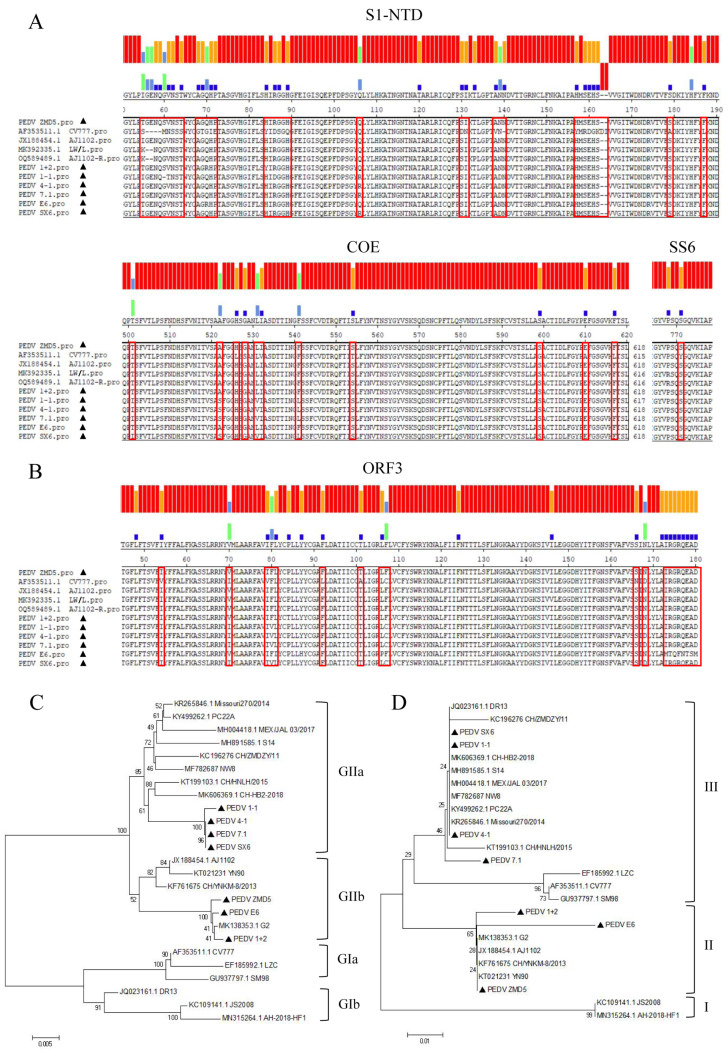
Sequence alignment and phylogenetic tree of S and ORF3 genes. (**A**) S1-NTD, COE, and SS6 sequence alignment, (**B**) ORF3 sequence alignment, (**C**) S Gene evolutionary tree and (**D**) ORF3 gene evolutionary tree. Sequencing samples are marked with black triangles in the diagrams.

**Figure 3 animals-13-01766-f003:**
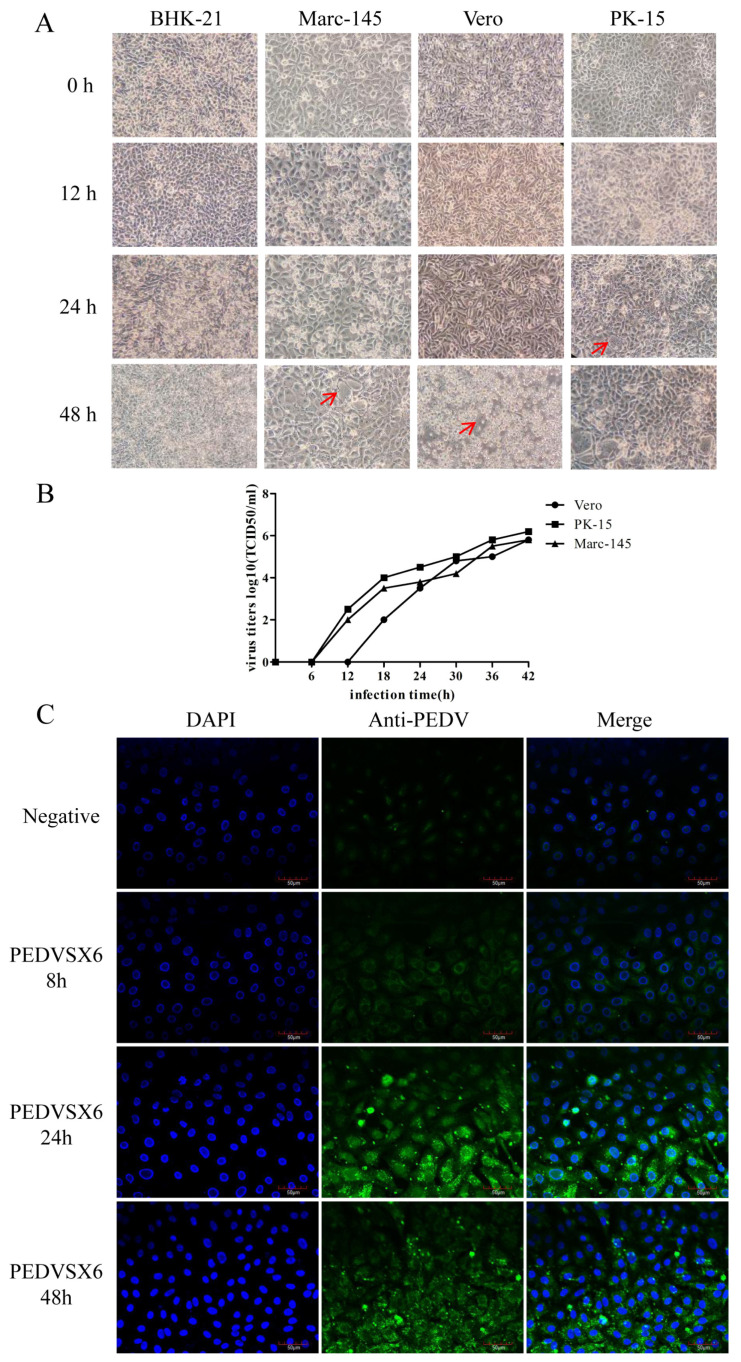
Infection and identification of isolates. (**A**) Infection of the isolates in different cells. Cellular changes were observed at 0, 12, 24 and 48 h. BHK21 cells were not infected with PEDV, and Marc-145, Vero and PK-15 cells were infected with PEDV and developed CPE as indicated by red arrows. (**B**) Growth kinetic curves of isolates infected with PK-15, Marc-145 and Vero cells. Culture supernatants were collected at 6, 12, 18, 24, 30, 36 and 42 h. The virus could hardly be detected in the culture supernatant at the early stage of virus invasion into the cells. The CPE increased gradually with time. Data are presented as mean ± SD by triplicates. (**C**) Indirect immunofluorescence validation of isolates. PEDV SX6 was the isolate, and negative was the control (the PEDV multiple antibody serum PE-S-20220302mp, negative serum PE-S-20220302mn). DAPI-stained nuclei, anti-PEDV was the polyclonal antibody of PEDV.A small amount of fluorescent signal was detected at 8 h post-infection with 0.1 MOI and gradually increased at 24 and 48 h.

**Figure 4 animals-13-01766-f004:**
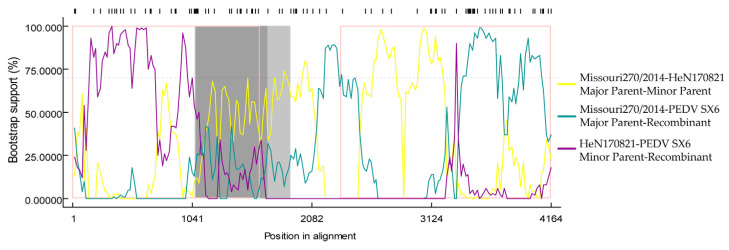
Prediction of recombination of S gene. RDP4 software was used to detect the possible recombination events of S gene. The potential breakpoints were 1071 and 4093 nt (reference sequence number: MK862249.1 and KR265846.1).

**Table 1 animals-13-01766-t001:** Sample collection information.

Name of Sample	Time of Collection	PEDV Vaccination Status (Sows)	PEDV Positive or Not
HB-lf-01	November 2019	Inactivated vaccine CV777	Negative
HB-lf-02	November 2019	Inactivated vaccine CV777	Negative
HB-lf-03	November 2019	Inactivated vaccine CV777	Negative
Ll	November 2019	Duplex live attenuated vaccine AJ1102	Negative
ZMD1	November 2019	Live attenuated triple vaccine CV777	Negative
ZMD2	November 2019	Live attenuated triple vaccine CV777	Negative
E1-4	December 2019	Inactivated vaccine CV777	Negative
E5	December 2019	Inactivated vaccine CV777	Negative
E6	December 2019	Inactivated vaccine CV777	Positive
1-1	December 2019	Inactivated vaccine CHYJ	Positive
2-1	December 2019	Inactivated vaccine CHYJ	Negative
3-1	December 2019	Inactivated vaccine CHYJ	Negative
4-1	December 2019	Inactivated vaccine CHYJ	Negative
5-1	December 2019	Inactivated vaccine CHYJ	Positive
6-1	December 2019	Inactivated vaccine CHYJ	Negative
1 + 2	December 2019	Duplex live attenuated vaccine LW/L	Positive
JZ-p1	December 2019	Live attenuated triple vaccine CV777	Negative
JZ-p2	December 2019	Live attenuated triple vaccine CV777	Negative
7.1	January 2020	Inactivated vaccine CV777	Positive
7.2	January 2020	Inactivated vaccine CV777	Positive
7.3	January 2020	Inactivated vaccine CV777	Negative
ZMD3	January 2020	Live attenuated triple vaccine CV777	Positive
ZMD4	January 2020	Live attenuated triple vaccine CV777	Positive
ZMD5	January 2020	Live attenuated triple vaccine CV777	Positive
C1	January 2020	Duplex live attenuated vaccine SCSZ-1	Positive
C2	January 2020	Duplex live attenuated vaccine SCSZ-1	Negative
C3	January 2020	Duplex live attenuated vaccine SCSZ-1	Negative
C4	January 2020	Duplex live attenuated vaccine SCSZ-1	Negative
4-1	November 2020	Duplex live attenuated vaccine AJ1102	Positive
3-2	November 2020	Duplex live attenuated vaccine AJ1102	Negative
2-2	November 2020	Duplex live attenuated vaccine AJ1102	Negative
1-1	November 2020	Duplex live attenuated vaccine AJ1102	Negative
E7-8	November 2020	Inactivated vaccine CV777	Negative
E9	November 2020	Inactivated vaccine CV777	Positive
HN-01	December 2020	Inactivated vaccine CV777	Negative
HN-02	December 2020	Inactivated vaccine CV777	Positive
HN-03	December 2020	Inactivated vaccine CV777	Positive
HN-04	December 2020	Inactivated vaccine CV777	Positive
HN-05	December 2020	Inactivated vaccine CV777	Negative
J1	December 2020	Live attenuated triple vaccine CV777	Negative
J2	December 2020	Live attenuated triple vaccine CV777	Negative
J3	December 2020	Live attenuated triple vaccine CV777	Positive
SX1-3	March 2021	Duplex live attenuated vaccine AJ1102	Negative
SX4-5	March 2021	Duplex live attenuated vaccine AJ1102	Positive
SX6	March 2021	Duplex live attenuated vaccine AJ1102	Positive
SX7-9	April 2021	Duplex live attenuated vaccine AJ1102	Negative
SX10	April 2021	Duplex live attenuated vaccine AJ1102	Positive

**Table 2 animals-13-01766-t002:** Reference sequences of virus strains.

Strain	GenBank Entry Number	Origin Country	Collection Year
CV777	AF353511.1	Belgium	1978
LZC	EF185992.1	China	2006
DR13	JQ023161.1	Korea	1999
AJ1102	JX188454.1	China	2012
PC22A	KY499262.1	USA	2017
Missouri270/2014	KR265846.1	USA	2016
S14	MH891585.1	Korea	2019
MEX/JAL/03/2017	MH004418.1	Mexico	2017
AH-2018-HF1	MN315264.1	China	2018
G2	MK138353.1	China	2018
CH-HB2-2018	MK606369.1	China	2018
CH/HNLH/2015	KT199103.1	China	2015
SM98	GU937797.1	Korea	2011
JS2008	KC109141.1	China	2013
YN90	KT021231	China	2015
CH/ZMDZY/11	KC196276	China	2013
NW8	MF782687	China	2017
CH/YNKM-8/2013	KF761675	China	2013

## Data Availability

Not applicable.

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
