# Peer review of "Natural Evolution of Porcine Epidemic Diarrhea Viruses Isolated from Maternally Immunized Piglets"

_animals, 2023, doi:10.3390/ani13111766_

Round 1

Reviewer 1 Report

Porcine deltacoronavirus (PDCoV) also cause gastroenteritis in pigs and is prevalent in China. The authors tested the clinical samples for PEDV, TGEV, and PoRV, but not PDCoV. The authors need to discuss this.

Since the primers were designed in this study, the authors need to provide additional information on which genes the primers target. Were the primers specific and sensitive? Figure 1B, there was no 299 bp-band in the P (controls) post sample # 33, suggesting that the assay was not sensitive for TGEV. This could be the reason for no TGEV was detected from this study.

It would be very useful to provide a table to show the farm vaccination status (live attenuated or inactivated PEDV vaccines) and PEDV detection rate.

Specific comments:

1.    Please spell out TGEV and PoRV when they appear in the paper for the first time.

2.    Abstract, the first sentence “Porcine epidemic diarrhea (PED) viruses can cause severe piglet mortality and immune failure in some herds.” Please revise it because PEDV does not cause immune failure.

3.    Lines 47-49. Please revise this sentence to “PEDV is a α-coronavirus and has a genome of about 28 kb in length, which consists of seven open reading frames (ORFs)“.

4.    Line 94. Please change “broth” to “medium”.

5.    Lines 108-109. The meaning of “then purified for three generations” is unclear.

6.    Line 130. Please provide the reference for Reed-Muench method.

7.    Please change “cytopathic lesions” to “cytopathic effects” (or CPE) throughout the manuscript.

8.    Line 226. Please change “virulence” to “CPE”.

9.    Fig. 2C legend. Please add cell line information. Which PEDV protein does the polyclonal antibody target? Please provide the source of this anti-PEDV antiserum (Line 143).

Reviewer 2 Report

The paper is on the characterization of PEDV isolates originated from maternally immunized piglets. Unfortunately, the work is not legibly written and needs general improvement. In its present form, it is not suitable for publication. In addition, there are many similar publications mainly by Chinese authors.

My comments:

1.      How were the animals immunized? Is there any confirmation of this (serological tests)?

2.      The introduction lacks information about the vaccines used in China, especially in Shanxi, Henan and Hebei provinces.

3.      Line 72-74. “Small intestine…..were collected from piglets that were infected or had died within 10 days of age….” Please revise the sentence. The word infected suggests that animals were infected experimentally. Rather, it should be written that the samples were from animals with clinical signs (if this was true). There is also lack of information, as the title of the paper indicates, that the animals were maternally immunized.

4.      Clarify the meaning of all abbreviations.

5.      Line 120. trypsin instead of enzyme

6.      Line 160-161. “Sequence splicing was performed using bioinformatics primer 5, DNAman, and DNAstar software as shown in Table 1.” I don't understand the sentence. Table 1 shows the reference strains used for phylogenetic analysis as I guess.

7.      The entire results section should be revised so that it becomes clear which and how many samples were tested with the given methods.

8.      Figure 1B. Not in all photos is a positive control and I have the impression that different markers were used because the bands are at different heights relative to the marker. Please revise and correct this.

9.      Figure descriptions should be next to the figure's captions and not treated as regular text.

10.  Line199-201. Requires clarification and correction. What kind of screening study was used? 18 of 20 samples positive in the RT-PCR test? I do not understand how all the samples were used to obtain a new strain of PEDV SX6.

11.  Line 207. What mean that the activity of Vero cells infected by virus disappeared? In how many samples was the cytopathic effect observed?

12.  Line 211. “The virus showed rapid proliferation in PK-15 cells.” The Figure2B does not show a significant differences between the different cells used.

13.  Are the results in section 3.2 related to one isolate PEDV SX6 or to the 18 positive PEDV samples? it should be clearly described.

14.  Section 3.3. Figure 2A, B, C, D. It is not clear which samples are reference and which are test samples. Test samples should be differently labeled from reference samples. SM95 and AM 2018-HF1 are reference or test samples? 20 samples were positive in the RT-PCR test. Sequences were not obtained for all of them?

15.  Line 248. Which tested isolates belonged to the GII group?

16.  “The nucleotide sequence homology accounted for 92.2%–98.4%.” Homology of what sequences?

17.  In reference to ORF3 sequence, nucleotide homology was 93.4%–100% and belonged to the same group III as PC22A. Homology of what sequences?

18.  To which samples do the recombination results refer?

19.  Line 288-289. “Meanwhile, the isolates can cause low titer infection in PK-15, Marc-145, and Vero cells.” What were the titers of the samples tested in this work? this information was not provided by the authors. What are the high or low titers? there is no comparison to the existing literature.

20.  Line 297-298. “Compared with that in the classical CV777 strain, the insertion or deletion of amino acid sequence mainly occurred in S1-NTD.” Is strain CV777 used to develop the vaccine in China in tested provinces?

21.  Line 300. “The mutation of S2-CTD was higher than that of S2-NTD.” Why do the authors say that?

22.  Line 309-310. “The ORF3 amino acid sequence of this isolate was intact and only mutated at AA21, 54, 79, 80, 92, and 101 compared with the classical strain.” Which classic strain?

23.  The results lack information on whether or not there is a deletion (49 or 51 nt) in the OFR3 sequences of the samples tested and what effect this has on the results observed in cell cultures.

24.  Conclusions should be revised. They are not based on the findings described in the paper

25.  There are many papers describing PEDV isolates from herds where the vaccine was applied. The authors should refer to the results of such studies.

Round 2

Reviewer 1 Report

Two major concerns:

1.    Figure 1B. The P (vaccine strain) had different number and sizes of bands in (1) from those in (4), and (6). Also, it looks like that the potentially positive bands in (4), (5) and (6) were smaller than 500 bp. Where are the positive control for TGEV and PoRV? Please explain these.

2.    Lines 378-379. There is no report that PEDV is a zoonotic pathogen. What does this sentence mean: “The continued emergence of new strains of PEDV not only causes significant 378 economic losses but also poses a potential threat to public health safety.”?

More specific comments:

Line 21. Delete “virus” in front of “(PED)”.

Line 49. “alphacoronavirus” should be “Alphacoronavirus” (capitalized and italic).

Lines 64-69. “diploid” and “triploid” should be “divalent” and “trivalent” for vaccines. Please indicate the genetic clusters (GIa, Gib, GIIa, or GIIb) for individual PEDV strains.

Line 95. “diethylpyrocarbonate (DEPC)” should be “DEPC-treated water”.

Lines 116-118. Why did the authors added this sentence here?

Line 156. “phage” should be “plaque”.

Lines 178-179. Please add the reference for Reed-Muench method, and delete” For the method, please refer to Appendix 37 of the 2020 edition of the Veterinary Pharmacopoeia of the People's Republic of China, Part III.”.

Reviewer 2 Report

Unfortunately, not all comments have been corrected by the authors. The paper still needs improvement. 

My comments:

1. Pictures are not very clear and, most importantly, should be larger.

2. Unfortunately, it is still not known whether the same vaccine was used in all herds or not (and it is not a question of the type of vaccine, but on the basis of which reference strain this vaccine was obtained). Table 1 shows that there were different vaccines. This is a very important aspect because the sequences obtained in this study should be compared to the reference strains used to develop the vaccines.

3. Line 227 The instead the

4. Line 279-281. Unfortunately, it is still not known which sequence comparisons apply to homology. This should be clearly written in the text. The authors used 18 reference strains to create phylogenetic trees while only four reference strains were used for sequence comparison. For comparison of aa sequences should be used strains on the basis of which the vaccines used in the study flocks were developed. 

5. line 291-293. It should clearly stated that the only strain that produced CPE was SX6. Please revise the text.

6. It should be clearly noted that recombination was detected in the SX6 strain. It is still unknown whether all 7 sequences were analyzed or only SX6. This requires correction. 

7. The materials and methods section lacks information on how many herds the samples came from. 

8. There is no information on what basis the 7 samples selected for sequencing were chosen. How many herds did these selected samples come from?

9. The discussion was not enriched by the results of similar work conducted in China and and conclusions have not been revised
